# Synergistic Action of Mild Heat and Essential Oil Treatments on Culturability and Viability of *Escherichia coli* ATCC 25922 Tested In Vitro and in Fruit Juice

**DOI:** 10.3390/foods11111615

**Published:** 2022-05-30

**Authors:** Luciana Di Gregorio, Alex Tchuenchieu, Valeria Poscente, Stefania Arioli, Antonella Del Fiore, Manuela Costanzo, Debora Giorgi, Sergio Lucretti, Annamaria Bevivino

**Affiliations:** 1Department for Sustainability, Biotechnologies and Agroindustry Division, ENEA, Italian National Agency for New Technologies, Energy and Sustainable Economic Development, Casaccia Research Center, 00123 Rome, Italy; luciana.digregorio@enea.it (L.D.G.); valeria.poscente@unitus.it (V.P.); antonella.delfiore@enea.it (A.D.F.); manuela.costanzo@enea.it (M.C.); debora.giorgi@enea.it (D.G.); sergio.lucretti@enea.it (S.L.); 2Centre for Food and Nutrition Research, Institute of Medical Research and Medicinal Plants Studies (IMPM), Yaoundé P.O. Box 6163, Cameroon; 3Food Evolution Research Laboratory, School of Tourism and Hospitality, College of Business and Economics, University of Johannesburg, Johannesburg 2000, South Africa; 4Department of Agriculture and Forest Sciences, Università degli Studi della Tuscia, 01100 Viterbo, Italy; 5Department of Food, Environmental and Nutritional Science, Università degli Studi di Milano, 20122 Milan, Italy; stefania.arioli@unimi.it

**Keywords:** food safety, shelf-life, mild heat treatment, *Origanum* essential oil, fruit juice, *E. coli*

## Abstract

The strengthening effect of a mild temperature treatment on the antimicrobial efficacy of essential oils has been widely reported, often leading to an underestimation or a misinterpretation of the product’s microbial status. In the present study, both a traditional culture-based method and Flow Cytometry (FCM) were applied to monitor the individual or combined effect of *Origanum* *vulgare* essential oil (OEO) and mild heat treatment on the culturability and viability of *Escherichia coli* in a conventional culture medium and in a fruit juice challenge test. The results obtained in the culture medium showed bacterial inactivation with an increasing treatment temperature (55 °C, 60 °C, 65 °C), highlighting an overestimation of the dead population using the culture-based method; in fact, when the FCM method was applied, the prevalence of injured bacterial cells in a viable but non-culturable (VBNC) state was observed. When commercial fruit juice with a pH of 3.8 and buffered at pH 7.0 was inoculated with *E. coli* ATCC 25922, a bactericidal action of OEO and a higher efficiency of the mild heat at 65 °C for 5′ combined with OEO were found. Overall, the combination of mild heat and OEO treatment represents a promising antimicrobial alternative to improve the safety of fruit juice.

## 1. Introduction

Alternative food processing and preservation technologies have been explored extensively in recent years to develop products with an extended shelf-life, as well as to preserve their nutritional and organoleptic characteristics in agreement with changing consumer demand [1]. The perishability and stability of food products over the course of their shelf life depends on both their intrinsic properties (pH, activity water -a_w_-, oxidoreduction potential) and extrinsic ones (storage temperature, gaseous atmosphere, and relative humidity) [2]. These factors influence the survival and growth of both spoilage and pathogenic microorganisms. Unlike the spoilage microorganisms, pathogens that are present in low levels may not produce noticeable changes in the food appearance. Hence, microbial growth control is essential to minimize foodborne diseases [3].

Recent interest in fresh fruit juices has induced researchers to develop alternative preservation methods which can maintain the nutritional compounds (e.g., vitamins, minerals, pigments, antioxidants, bioactive compounds) for longer periods [4,5,6,7]. Fruit juice plays an important role in daily nutrition as it is rich in antioxidant and beneficial compounds. The industrial process of pasteurization and aseptic filling allow the stabilization and preservation of the fruit juice at room temperature; however, the intensity of thermal treatment can negatively affect the nutritional and organoleptic characteristics of the final product [1,8]. To prevent spoilage and preserve the freshness of the fruit juice, several processing technologies have been applied, such as ohmic heating, microwave heating, thermosonication, pulsed electric fields, ultraviolet irradiation, high hydrostatic pressure, and pulsed electric field [9,10]. However, most of these processes require cutting edge technology and high financial investments, representing a significant barrier especially for many small and medium enterprises [11,12]. In addition, their individual activity can show only a slight efficacy on microorganism inactivation or even a negative effect on the physicochemical characteristics of the juice [10,13,14,15].

During the last decade, a combination of antimicrobial compounds—such as essential oils (EOs)—and other preservation technologies, such as low temperature treatments, have been proposed for fruit juices that are stored at room temperature, in order to reduce the survival and growth of both spoilage and pathogenic microorganisms [16,17,18,19,20,21]. EOs are “generally recognized as safe” (GRAS) food additives with antimicrobial properties and are approved by the USFDA for their use in foods and drinks (USFDA 2015). Essential oils of different *Origanum* species have exhibited good antimicrobial effects against several bacteria, such as *Staphylococcus aureus*, *Salmonella* spp., *Pseudomonas* spp., and *Escherichia coli*, and fungi including *Candida albicans* and *Aspergillus* spp. It was shown that oregano (*Origanum vulgare*) and thyme (*Thymus vulgaris*: light and red varieties) EOs had the strongest bacteriostatic and bactericidal properties against *E. coli* strains [22]. The *Origanum* essential oil is known to be rich in compounds like thymol, carvacrol, and γ-terpinene, as well as *cis*- and *trans*-sabinene hydrate, with effective antimicrobial activity [23]. The characteristic hydrophobicity of EOs enables them to penetrate the cell membrane, determining the leakage of cell content [24]. On the other hand, mild heat treatment can enhance the antimicrobial activity of EOs by promoting the formation of the vapor pressures of the volatile organic compounds, which, in turn, increases their solubility in the cell membrane, the first target of their antimicrobial activity [13]. The efficacy of EOs combined with mild heat treatment can represent a safe, cost-effective compromise, minimizing the damages caused by high temperatures. 

Despite the recent progress in elucidating the synergistic activity of EOs and mild heat treatment [7,13,16,17,25,26,27,28,29,30], little is known about the effect of antimicrobial-assisted pasteurization of fruit juice on food pathogen viability [21,31,32]. It is well known that the use of traditional, culture-based microbiological approaches can lead to an overestimation of treatment efficacy [33]. Indeed, foodborne bacteria, induced by stressful conditions, such as low temperature, pH, high osmolarity, and nutrient starvation can enter into a particular physiological state named “viable but nonculturable (VBNC)”. The main characteristic of VBNC cells is their non-cultivability. These cells are not able to grow and replicate on standard solid culture media, eluding detection by using conventional microbial culture-based techniques. VBNC bacteria cells exhibit a state of dormancy or a basal metabolic activity but, being in this state reversible under certain conditions, they can “resuscitate”, keeping their pathogenic potential and regaining virulence in the environment and therefore posing a serious threat to food safety and public health [34,35,36].

*E. coli* represents a threat to global public health as it is implicated in many food-borne outbreaks, especially those that are associated with unpasteurized fruit juices, including apple and orange juice, but also to fermented apple juice, and cider [13,37]. EOs addition and mild heat treatment can inactivate and induce VBNC *E. coli* during food processing [38]. Therefore, the evaluation of the ratio of death and VBNC cells that is related to the applied treatment represents a crucial point to ensure food safety, in order to avoid the processing conditions that are suitable for the induction of VBNC *E. coli* [39,40]. An innovative approach to assess both the culturability and viability of food pathogens in complex matrices such as juice fruit is needed to optimize the protocol used for food production and evaluate the effectiveness of preservation treatments. Among the culture-independent techniques, flow cytometry (FCM) has been displayed to be a valid and sensitive tool for the fast analysis of bacterial populations, providing unique information on bacterial viability and physiology [41,42,43]. Moreover, flow cell sorting (FACS) associated with FCM allows the physical isolation of specific subpopulations, characterizing their physiological state. This single-cell approach offers the opportunity to estimate the VBNC “resuscitation” phenomenon and, consequently, the specific food preservation method’s effectiveness. In reality, most of the available food-related studies use FCM for liquid samples, including fruit juice, for a rapid and reliable detection of foodborne pathogens in the food industry, in order to minimize foodborne diseases [41]. In the present work, we aimed at exploiting the synergistic effect of *Origanum* EO and mild heat treatment on the *E. coli* inactivation in a conventional culture medium (in vitro test), and in a fruit juice challenge test, evaluating both culturability and viability following the combined treatments.

## 2. Materials and Methods

### 2.1. Bacterial Strain and Inoculum Preparation

The microorganism used in this work was the non-pathogenic *E. coli* ATCC 25922 strain. This strain was used as a reference strain and chosen for its ability to survive at relative low pH values, as previously stated by Tchuenchieu et al. [17]. The stock culture was stored at −80 °C in 15% (*v*/*v*) glycerol for further use. A sample was sub-cultured into 5 mL of fresh Brain Heart Infusion (BHI) medium (Merck KGaA, Darmstadt, Germany) and incubated without shaking for 24 h at 37 °C. Then, an aliquot of 1 mL was transferred into 9 mL of fresh BHI medium. After 24 h of incubation at 37 °C, 1 mL was transferred into 30 mL of fresh medium and incubated at 37 °C for 24 h. Starting from the refreshed culture, the *E. coli* ATCC 25922 initial concentration that was applied in antimicrobial treatments was 6 Log CFU/mL.

### 2.2. Bacterial Culturability and Viability Assays

The bacterial survival ratio of *E. coli* ATCC 25922 after different treatment conditions was estimated using both the culture-based method by plating procedure on MacConkey agar, and the FCM methods applying SYBR Green I/Propidium Iodide (PI) double-staining.

The effect of thermal treatment on the culturability and viability of *E. coli* was evaluated at 55, 60, and 65 °C, according to the procedure described by Tchuenchieu et al. [25]. Briefly, for each of the tested conditions, it consisted of a vial that was introduced in a thermostatically controlled water bath whose temperature was fixed to the desired temperature. Then, 1 mL of a tenfold dilution of the cell culture was inoculated into the vial containing 99 mL of Phosphate Buffer Solution (PBS, pH 7.4) that was already preheated at the desired temperature. The initial concentration was 6 Log cell/mL. The vial was then kept in the water bath for 30 min and immediately cooled by immersion in a cryogenic solution. The untreated *E. coli* in BHI was used as a positive control, and the *E. coli* sample in BHI autoclaved at 121 °C for 15 min as a negative one.

The effect of *Origanum* essential oil (OEO) alone and in combination with mild heat treatment was evaluated in the in vitro tests, in BHI medium. Different temperatures (55 °C for 30 min and 65 °C for 5 min), chosen based on the thermal treatments results as described above, were investigated. The 100% OEO [Farmalabor Srl, cod. 1686A, Canosa di Puglia (BT), Italy] was diluted in ethanol (absolute, ≥99.8%), to obtain stock solutions of 1% (*v*/*v*) and 0.5% (*v*/*v*), then filtered through a 0.2 μm pore size polycarbonate membrane [21]. Each tube for each treatment contained 9.8 mL of BHI; 0.1 mL of 6 Log cell/mL bacterial culture; and 0.1 mL of OEO, 1% or 0.5%, achieving a final concentration of 100 and 50 ppm, respectively [16]. Three biological replicates were prepared for each tested condition and analyzed immediately after inoculation (T = 0), and after 4, and 24 h of incubation at 37 °C. The positive control was the not-thermally treated *E. coli* in BHI and the negative one was the *E. coli* sample in BHI autoclaved at 121 °C for 15 min which was also used.

The juice fruit challenge test was performed using a commercial fruit juice [“Tantifrutti” apricot, peaches and apple with basil) produced by the company Rigoni di Asiago Srl, Asiago (VI), Italy]. The juice was first centrifuged at 8000× *g* for 10 min and 4 °C to reduce the aspecific background; the supernatant was used for the experiments, and the pH value was determined using a portable pH meter [edge^®^ pH/ORP, Hanna Instruments Italia Srl, Villafranca Padovana (PD), Italy]. The fruit juice was artificially contaminated with a concentration of 5 Log cell/mL of *E. coli* ATCC 25922 strain in two different conditions: at its own pH 3.8, and a pH adjusted up to 7 with NaOH 0.5 M. As a positive control, *E. coli* grown in BHI medium was used. A combination of mild heat treatment (65 °C for 5 min) and OEO at 50 and 100 ppm was applied. The bacterial survival ratio at room temperature (25 °C) was estimated by using culture based and FCM methods. 

#### 2.2.1. Standard Plate Count Method

One-hundred microliters of serial 10-fold dilutions in a saline solution (NaCl 0.9% *w*/*v*) of bacterial suspensions from in vitro and fresh fruit juice experiments were plated on MacConkey agar. The colonies were counted after 24 h of incubation at 37 °C and 25 °C, for the in vitro experiments and juice fruit challenge test respectively. Final data, given as log CFU/mL (Colony Formation Unit: CFU), resulted from at least three independent experiments with three replicates each.

#### 2.2.2. Flow Cytometry Analysis (FCM)

FCM analyses were carried out using a flow analyzer CytoFLEX S (Beckman Coulter, Flow Cytometry, Milan, Italy), and a flow cell sorter FACS Vantage SE (Becton Dickinson, Biosciences Unit, Milan, Italy). Regions of interest were defined in relation to the positive and negative controls using a double-staining procedure [SYBR Green I (using a 1:10,000 dilution of the stock reagent) and Propidium Iodide (PI) 10 µg/mL] [21]. The viable cells exhibited a green signal (SYBR positive) while the damaged/dead cells showed an orange-red fluorescence (PI positive). Subpopulations (viable, damaged, and dead populations) were marked on an SYBR Green I vs. PI dot plot according to a different green/red fluorescence ratio, and their relative positions were delimited on all subsequent FCM dot plot and density plot analyses [21]. FCM samples were prepared with a one-hundredfold dilution of the samples in filtered (0.2 µm) PBS (pH 7.4), double-stained, distributed in 96-well plates, and incubated for 15 min at 37 °C before use. The cell suspensions were analyzed by using the blue laser (ext. 488 nm) and the bandpass filters collecting fluorescence emissions at BP525/40 and BP675/50 for the SYBR Green I and PI fluorescence, respectively. Acquisition of 30,000 events per sample was carried on in 3 replicates with a medium flow rate (30 µL/min). The signal trigger was set on a dual-parameters mode as Side Scatter SSC, which approximates the particle shape and texture vs. PI red fluorescence (PI sensor gain set at background intensity levels), in order to allow a reduction in non-specific signals generated by very small debris during the analysis. Microspheres of 2.5 µm in diameter (Alignflow™ for Blue Lasers, Thermo Fisher Scientific Life Science Solutions, Milan, Italy) were used as an internal reference standard in each sample. The parameters were acquired with a logarithmic scale and analyzed using the CytExpert software v. 2.3 (Beckman Coulter Flow Cytometry, Milan, Italy)). Flow cell sorting was carried out through a FACS VANTAGE SE (Beckton Dickinson, Biosciences Unit, Milan, Italy) equipped with a 488 nm ext. laser at 400 mW power output using a 70 µm ceramic nozzle and 20 psi sheath fluid pressure. The fluorescence was collected through a band pass 530/30 nm filter for the SYBR Green I and a long pass 620 nm filter for PI, and analyzed by a CellQUEST v. 4.1 (Becton Dickinson, Biosciences Unit, Milan, Italy). The cells that were identified by each subpopulation were automatically distributed in 96-well plates filled with MacConkey Broth culture medium at concentrations of 1, 10 and 100 cells/well using a computer-controlled cell deposition unit (ACDU, Becton Dickinson, Biosciences Unit, Milan, Italy). Growth was evaluated by measuring the OD at 595 nm using a microplate-based assay (VICTOR plate reader, PerkinElmer, Milan, Italy).

### 2.3. Statistical Analysis

The statistical analysis was performed by using Past4©. Data are presented as mean ± Standard Deviations (SD) based on triplicates from at least three independent experiments. Data were compared using several-sample tests of ANOVA, with Tukey’s pairwise test at *p* < 0.05 considered statistically significant (95% confidence interval).

## 3. Results and Discussion

The effect of OEO and mild heat treatment on the culturability and viability of *E. coli* ATCC 25922 was assessed in conventional culture medium and inoculated fruit juice through culture-based methods and FCM. In a previous work [17], the use of a natural antimicrobial compound such as carvacrol [44] that is present in high proportions in *Origanum* spp. essential oil [45] was found to increase the efficiency of mild heat treatments and to counteract the enhanced thermal resistance of acid-adapted *E. coli* ATCC 25922 cells. During the last decade, many researchers demonstrated the potentiating effect of a mild temperature treatment on the antimicrobial efficacy of essential oils and microbial inactivation [18,26].

In the current work, we aimed at investigating the effect of the use of mild thermal in combination with OEO treatment on both the culturability on BHI medium and the viability assessed by flow cytometry. The combination of both detection methods was applied to determine the proportions of the different viability states of bacteria and to highlight discrepancies between the viability and culturability data. SYBRGreen I/Propidium Iodide (PI) double-staining was used to differentiate the subpopulations (dead, live, damaged cells), since the SYBR Green I stains the nucleic acids of all cells to produce green fluorescence, while the PI is a membrane-impermeant dye that intercalates into nucleic acids of dead or damaged cells to generate red fluorescence [46]. Bacteria that is characterized by a certain degree of injured membrane presents staining characteristics of both dye emissions, according to the amount of PI able to penetrate the cells, and depending on the significance of the membrane damage [24]. Single-cell data generated by FCM analyses allow the defining of signals that are measured as Total Fluorescent Units (TFU), including all stained cells emitting fluorescence, and embracing viable, dead, and damaged ones. The ability to discriminate viable cells from damaged and dead cells enables a way of numbering viable cells as Active Fluorescent Units (AFU: total cell number minus damaged and dead ones), thus complementing plate counting (CFU) and membrane integrity (AFU) via FCM measurements [47].

### 3.1. Effect of Thermal Treatments on E. coli ATCC 25922

Under the heat effect, bacteria cells are inactivated, as their DNA, ribosomes, proteins, and enzymes are affected, and their outer layers and membrane are damaged, depending on the severity of the treatment [48]. The *E. coli* ATCC 25922 inactivation and evaluated by a culture-based method clearly showed an overestimation of the dead population as the treatment temperature increased, compared to FCM methods. The plate count method merely enumerates those cells that can replicate under the specific conditions, wrongly assuming that all the others are dead. Thermal treatments can trigger the occurrence of VBNC populations, which were stressed and lost their ability to grow on agar medium [43]. Therefore, although bacteria cells still presented metabolic activity, the quantification of VBNC bacteria was not possible with conventional plating [49]. The FCM method detects cells and gives information on bacterial load and those dead or under sublethal conditions, despite their ability to survive. Unlike the time-consuming and labor-intensive standard methods (culture and PCR techniques), FCM provides a quick response and the possibility to observe the injured bacterial subpopulations and evaluate the presence of VBNC [50,51]. In the present study, a decrease in the culturability of >99% was observed after treatment at 55 °C for 30 min. Interestingly, 78% of the cells resulted as alive by FCM analysis (Table 1). 

The discrepancy between culturability and viability was also observed in the other conditions (60 °C and 65 °C), therefore suggesting that a high fraction of treated cells remained alive but became non-culturable after exposure to heat treatment (Figure 1). In fact, exposure to unfavorable environmental conditions, including extreme temperature, oxidative stress, acid stress, and nutritional starvation had already been reported as inducing a VBNC state in bacteria [52,53].

The running of PBS that was inoculated with stained *E. coli* culture on FCM enabled definition of the region corresponding to this microorganism live population. Based on this and on the negative control, three regions were defined: P1, P2, and P3. It appeared that with the increase in treatment temperatures the population shifted from region P1 (alive cells) to P2, and then from P2 to P3 (Figure 1). Considering the 60 °C and 65 °C heat treatments, an increase in injured cells was observed (46% and 34%, respectively), in comparison with the 55°C treatment, suggesting a later potential cell recovery. Except for the not-thermally treated sample (positive control), a significant difference was noticed when evaluating alive cell content in each sample with both methods, what with flow cytometry being faster and more sensitive, considering the small standard error obtained.

### 3.2. Effect of OEO Treatment

The efficiency of OEO treatment was evaluated by comparing the culturability and viability, quantifying the bacterial cells that were able to replicate using plate counts, and extrapolating the percentage of cell viability that was obtained from the FCM analysis, representing the subpopulations with different levels of resistance to the antimicrobial agent. The *E. coli* cells were exposed to sub-inhibitory growth concentrations of OEO (50 and 100 ppm) [54]. Plate counts showed a significant decrease in CFU/mL for both 50 and 100 ppm, compared to the untreated sample (0 ppm) (*p* < 0.05) (Figure 2). The density plot displayed a shift in the PI region (Figure 3).

An immediate significant effect of 100 ppm OEO on viable cells was observed, with a decrease of 15% of alive cells and an increase of 13% of dead cells (Figure 4). The treatment efficacy with 100 ppm of essential oil was still evident after 4 h of incubation, corresponding to a significant decrease in viable cells by the FCM analysis (−14%; Figure 4). OEO antimicrobial activity could be attributed to its high ratio of phenolic compounds (carvacrol, thymol, *p*-cymene, and their precursor c-terpinene) [55]. Through their hydrophobicity, phenols exert an inhibitory effect, interacting with microorganisms’ cell membrane, altering its permeability, thus producing a loss of ions such as protons, phosphorus, and potassium. The loss of ions has tremendous effects on the proton motive force, thus reducing the content of intracellular ATP and compromising the total cell activity [56]. Nevertheless, after 24 h of incubation, no significant differences between treated and untreated samples were observed. A subsequent increase in the cells into the viable region was detected (Figure 3), suggesting a diminishing effect of OEO when *E. coli* grows in optimal conditions (pH 7.4). This is probably due to the hydrophobicity of EOs, which typically increases at lower pH values, allowing them to dissolve more easily into the cell membrane lipids of bacteria [13].

### 3.3. Effect of Antimicrobial Treatment in Synthetic Medium and Recovery Ability: Mild Heat-OEO

The effect of the 55 °C for 30 min thermal treatment, immediately after the inoculation, assessed by the culture-based analysis, determined a significant decrease in culturable cells compared to the untreated sample (0 ppm) (*p* < 0.05), whereas the combined treatment revealed no growth at the initial incubation time. A slight and total recovery after 4 and 24 h, respectively, was observed in all tested conditions (Figure 5). 

The FCM density plot presented a decrease in viable cells in thermal treatment combined with OEO at time 0 (−17% at 50 ppm and −42% at 100 ppm) and after 4 h of incubation (−25% and −35% at 50 ppm and 100 ppm, respectively). After 24 h, there was still a significant difference in terms of dead cells between thermal and combined treatment, with no significant variation among the oil concentration that was tested. However, a shift towards the alive region was found (Figure 6a and Figure 7a).

The treatment of OEO at 65 °C led to a clear reduction in the alive population by 47–50%, compared to the untreated sample (Figure 7b). Immediately after the incubation, a significant increase in injured cells in thermal treatment and a corresponding increase in dead cells in the combined treatment with OEO, 50 and 100 ppm was observed. A similar trend after 4 h of incubation was detected even if a shift towards the damaged region was highlighted. The growth recovery after 24 h is well pointed out in the dot plot (Figure 6b), whereas a complete loss of cell culturability for the combined mild heat treatment and OEO at 100 ppm was observed, validating the bacteriostatic effect of OEO [23] (Figure 5). Stressed cells constitute an “undetectable” population that are not able to grow on agar plates, therefore leaving an open question on the potential risks for real food matrices if suitable preservation and monitoring practices are not followed. Mild heat treatments increase the inhibitory effect of OEO by altering the membrane fluidity and composition and promoting the formation of the vapor phase of the oil’s volatile compounds [7,13]. OEO can coagulate the cytoplasm and determine damage to lipids and proteins, potentially leading to cell lysis [24]. Comparing the two mild heat treatments, the higher temperature determined more substantial damage to the cells, although the exposure times were lower, thus increasing the OEO effect. In fact, the temperature-induced cellular stress enhanced its bacteriostatic action even at a non-acid pH [13]. Moreover, during these combined treatments, the antimicrobial activity of OEO was possibly enhanced by its vapor pressure and its octanol/water partition coefficient (hydrophobicity) that varied with the tested treatment temperature [7].

### 3.4. In Vitro Treatments Efficacy Assessment: Subpopulations Culturability 

The flow cell sorting was performed on viable and dead cells (Appendix A: H1-LR viable and H1-UL dead, respectively) to verify the effectiveness of treatments on *E. coli* ATCC 25922 and the cloning capabilities. In the viable control, cells from the alive region showed a variable percentage of growth depending on cell plating density: 75%, 88%, and 100% of wells revealed colony formation by inoculating 1, 10, and 100 cells per well, respectively. No growth was observed in the inoculated wells that were sorted from the dead region (Appendix A; H1-UL), thus confirming the validity of flow sorting based on FCM data and the dual staining procedure for the counting of dead/inactivated cells. A decrease in the growth percentage by 10–25% and 25–37% was observed in the viable region (H1-LR) for the samples that were treated with a combined thermal treatment at 55 °C for 30 min and 65 °C for 5 min, respectively. In particular, for the samples that were subjected to OEO and mild heat thermal treatment, the cells that were identified by each subpopulation were also inoculated on MacConkey Agar plates according to a flow sorting matrix structure of 128 cells. The viable fraction was thus calculated as a percentage of colonies on the total numbers of cells inoculated per plate (Appendix A). The multiparametric FCM method, by staining different cellular targets, can be used to define the subpopulations of interest. Therefore, this single-cell approach produces data from a large population and provides a more complete profile of the internal population heterogeneity [57]. For the food industry, FCM offers the real-time analysis of a large number of cells from a small sample volume [58]. The application of the fluorescence-activated cell sorting (FACS) technique allows the isolation of specific FCM subpopulations for subsequent manipulations. FACS consents to evaluate the real treatment’s efficacy at a single-cell level, following the potential recovery of the sub-lethal damaged cells (on liquid or agar medium) after mild treatments [59,60]. The potential of an FCM analysis is its ability to reveal several levels of population heterogeneity that are induced from different degrees of cell damage not detectable by traditional methods [57,61]. The results obtained open the possibility to conduct additional investigations for correlating different methods’ data. FACS is a unique technique allowing the identification and sorting of single cell types according to molecular and physiological features, extracting diversity from heterogenic cellular populations. Thus, FACS represents a powerful tool to deeply investigate the relationship between FCM and traditional techniques, creating a strong link between different but complementary methodological approaches [62].

### 3.5. Effect of Antimicrobial Treatment in Fruit Juice and Recovery Ability: Mild Heat-OEO

#### 3.5.1. Microbial Challenge Test in Fruit Juice at pH 3.8 

When OEO alone was used, results revealed that at the inoculation time in all samples (0, 50, 100 ppm) the damaged subpopulation was predominant but not significantly different (*p* > 0.05), probably as a result of the low pH effect on *E. coli* growth. However, a significant biocidal effect of OEO 100 ppm was observed (+13% of dead cells population vs. untreated samples) (Figure 8a). 

After 24 and 48 h of incubation, a biocidal effect of OEO 50 ppm (+18% and +7% of dead cells at 24 h and 48 h) and 100 ppm (+24% and +26% of dead cells at 24 h and 48 h) was detected and a significant decreased of culturable cells was observed (*p* < 0.05), consistently (Figure 9a).

At 48 h of incubation, samples that were treated with 50 ppm of OEO displayed a higher percentage of injured cells (69%) compared to 100 ppm (48%), while the latter showed a similar percentage of damaged and dead cells (48% and 44%, respectively) (Figure 8a, Figure 10a). These results showed that the fruit juice’s low pH promoted the OEO’s hydrophobicity and, therefore, the OEO was able to carry out its biocidal activity. In this case, the consistency between the data obtained with plate count and FCM methods emphasized how FCM can reveal the real presence or absence of VBNC. Moreover, FCM analysis allowed us to quantify damaged cells from different sub-lethal OEO concentrations and evaluate the different efficacy ratios [21].

When mild heat-OEO treatment was used, results suggest that temperature affects alive cells, i.e., a significant decrease in alive cells at inoculation time up to 48 h of incubation compared to the untreated control was observed (*p* < 0.05). A high percentage of injured cells after 24 h of incubation was detected in all tested conditions. Figure 8b indicated a significant biocidal effect of OEO at 50 ppm and mild heat treatment as indicated by an increase in dead cells of 11% was detected at inoculation time, rising to 12% after 24 h of incubation, compared to the thermal treatment alone. As the oil concentration increased (100 ppm), there was an increment in dead cells of 12% and 20% (time 0 h and 24 h, respectively) (Figure 8b, Figure 10b). No bacterial growth by plating *E. coli* was observed in all conditions, confirming the presence of stressed subpopulations that are not able to grow on agar plates (Figure 9b). The synergy between mild thermal treatment and essential oils has already been widely demonstrated for several microorganisms [63]. Interestingly, a recent study found that the mild heat treatment when combined with thymol and carvacrol caused a partial or total inactivation of *Listeria monocytogenes*, mainly affecting cell culturability rather than viability. The culturability of the cells was also less affected when the inoculum derived from a single colony, indicating the importance of the physiological state of the cells that are used in an antimicrobial assay [32]. The discrepancy between the culturability and viability that was observed in the present study highlighted the overestimation of the treatment success if considering the classical plate counting. As for the results that were obtained in vitro, the FCM analysis provided information on the cell sub-populations’ distribution based on their physiological state, showing that most of the cells were damaged rather than dead [21,64].

#### 3.5.2. Microbial Challenge Test in Fruit Juice at pH 7.0 

OEO alone significantly affected bacterial growth up to 24 h. In particular, the treatment with 100 ppm OEO was more effective in reducing the bacterial load (−21%, and −10% of alive cells at 0 and 24 h of incubation, respectively), according to the results that were obtained in the in vitro experiment. The percentage of viable cells decreased in correlation with the essential oil concentration, showing, at the same time, a percentage increase in damaged and dead cells. After 48 h of incubation, the optimal growth conditions resulted in the bacterial growth recovery (85% and 89% of alive cells for 50 ppm and 100 ppm treatment, respectively), determining no significant differences between the treated and untreated samples (Figure 11 and Figure 12a).

The plate cell count displayed a similar trend, with a significant difference at the inoculation time (0 h) for the 100 ppm EO treatment (*p* < 0.05) (Figure 13). As for the in vitro results, the OEO treatment appeared to be less effective, probably due to the interaction with the bacterial membrane, inhibited by the neutral pH [24].

The thermal treatment at 65 °C for 5’ determined a significant reduction of 31% of the alive population compared to the untreated control (Figure 12b). When mild heat-OEO treatment was used, a complete loss of cell culturability was detected up to 48 h, suggesting the prolonged efficacy of this treatment compared to the thermal one (Figure 13b). Nevertheless, the high percentage of damaged cells population (between 49% and 65%) at all incubation times revealed the prevalence of injured cells, which was detectable only by the FCM analysis (Figure 11 and Figure 12b). The combined treatment proved to be more effective compared to the thermal treatment and OEO applied separately: the thermal treatment was effective up to 24 h, while together with OEO, the antimicrobial effect was observed up to 48 h. Indeed, mild heat is responsible for the first cellular damage step, an essential prerequisite for OEO antimicrobial activity on target cells. The membrane injuries increase permeability, causing the leakage of ions and molecules from the cell [22].

## 4. Conclusions

There is a need for alternative combined mild treatments that can preserve the safety of fruit juices and, at the same time, allow them to maintain the properties of “cold-pressed juices”. We found that mild heat treatment induced the VBNC state of *E. coli* ATCC 25922 while a strong reduction in VBNC induction was found when used in combination with OEO. The study highlighted the importance of combining traditional culture-based analytical methods with fast and high-throughput ones for monitoring the microbiological safety of fruit juice. The comparison (solely on an economic basis) between the classic microbiological technique and flow cytometry for food applications clearly works in favor of the classic technique, but FCM is a tool which extends microbiological analyses capabilities and has a huge number of applications in several fields [65,66,67], and diminishing costs and user-friendly software are increasingly promoting its use. Flow cytometry proved to be capable of providing information on a large number of cells during the different phases of their growth, exceeding the limitation of the traditional culture-based methods, which underestimated bacterial growth. Indeed, the use of FCM is essential as a real-time early warning monitoring system for preventing the risk of the spread of food-borne disease. In perspective, to avoid bacterial food contamination, it is important to perform a thorough investigation into the VBNC population, which presents potential recovery under certain favorable environmental conditions, even remaining non-culturable. The analysis of the VBNC induction will be extremely helpful in investigating its viability vs. culturability, and in deeply exploring the different factors that trigger the exit of the cells from the VBNC state. Further, the sorting of damaged cells could help in understanding how VBNC cells retain the capacity to recover their ability to replicate and cloning, and to verify the effects of antibiotic agents on controlling this competence.

In conclusion, the present research showed how mild heat treatment combined with essential oil can be a valuable alternative for food preservation technologies. Nowadays, finding alternative to the traditional methods is crucial for responding to the increasingly conscious consumer’s request for food with unchanged organoleptic properties that are as similar as possible to fresh products. Additional and more detailed studies should focus on fully understanding the thermal treatment combined with the natural aroma compounds’ mechanism of action for elaborating predictive models that are suitable for application in industrial processes.

## Figures and Tables

**Figure 1 foods-11-01615-f001:**
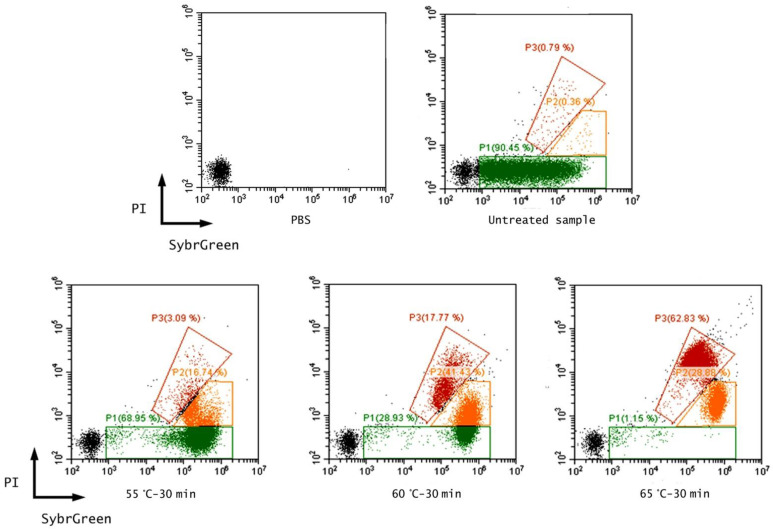
FCM dot plots obtained with a sample of PBS, PBS inoculated with *E. coli* ATCC 25922 (untreated sample), PBS inoculated with *E. coli* ATCC 25922 and treated at 55 °C, 60 °C, and 65 °C for 30 min.

**Figure 2 foods-11-01615-f002:**
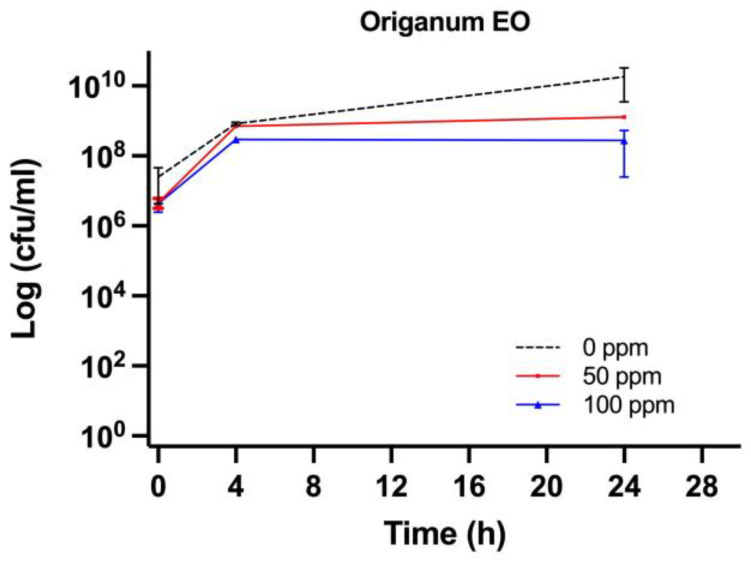
*E. coli* cell load in Brain Heart Infusion medium and plated on MacConkey agar (Log CFU/mL): OEO (0, 50, and 100 ppm).

**Figure 3 foods-11-01615-f003:**
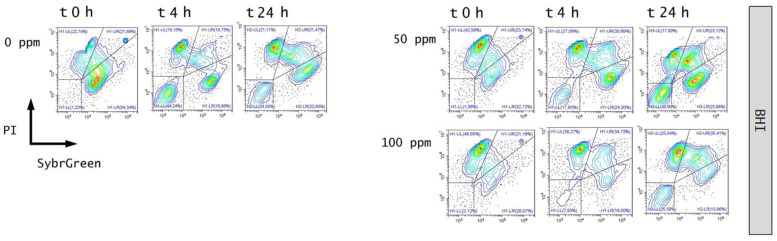
Double-staining density dot plot of *E. coli* cells in BHI broth diluted in PBS (pH 7.4): 50 ppm and 100 ppm essential oil treatment compared to 0 ppm (control) (at 0, 4 and 24 h of incubation). Cells were stained with SYBR Green I and PI simultaneously. H1-LL: unstained debris; H1-LR: intact cells/viable cells (SYBR Green I); H1-UR: injured cell population; H1-UL: permeabilized/dead cells (PI). In the upper right side of the plot, a defined single population represents standard beads.

**Figure 4 foods-11-01615-f004:**
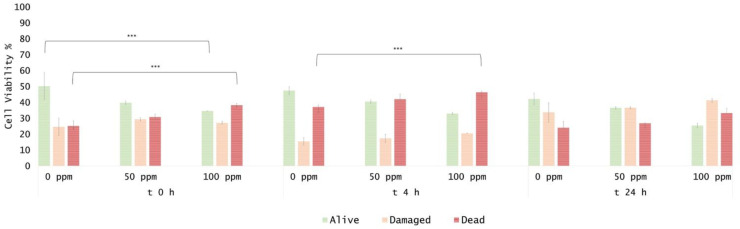
Percentage of alive, dead, and damaged cells calculated on total FCM events (100%): 50 ppm and 100 ppm OEO treatment at 0, 4, and 24 h compared to 0 ppm. The shown values correspond to the means and standard deviations for replicates, independently analysed. *p*-value less than 0.001 is flagged with three stars (***).

**Figure 5 foods-11-01615-f005:**
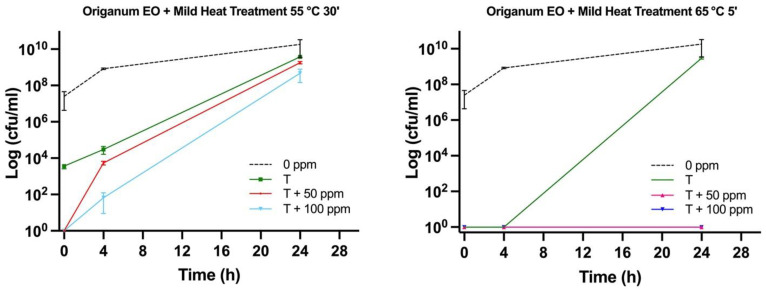
*E. coli* cell load in Brain Heart Infusion medium and plated on MacConkey agar (Log CFU/mL). Thermal treatment (55 °C 30 min) alone and combined with OEO (50 and 100 ppm); thermal treatment (65 °C 5 min) alone and combined with OEO (50 and 100 ppm).

**Figure 6 foods-11-01615-f006:**
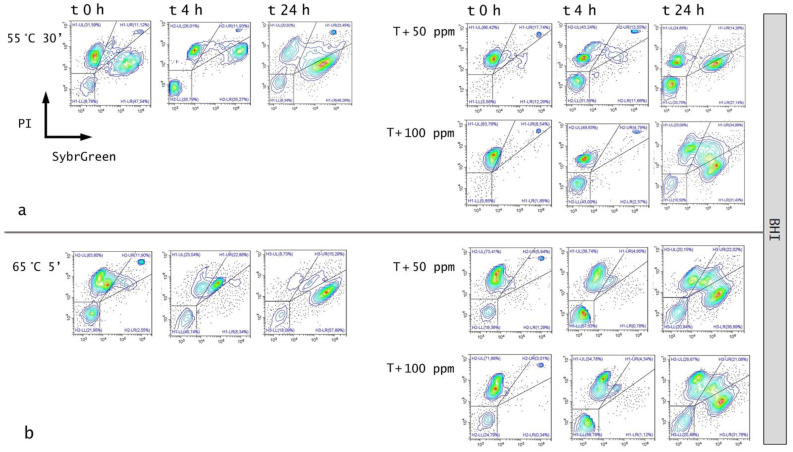
Double-staining density dot plot of *E. coli* cells in BHI broth diluted in PBS (pH 7.4): (**a**) combined thermal treatment at 55 °C 30’ and (**b**) 65 °C with or without OEO (50 and 100 ppm) at 0, 4, and 24 h of incubation, compared to thermal treatment alone. Cells were stained with SYBR-Green I and PI simultaneously. H1-LL: unstained debris; H1-LR: intact cells/viable cells (SYBR Green I); H1-UR: injured cell population; H1-UL: permeabilized/dead cells (PI). In the upper right side of the plot, a defined single population represents standard beads.

**Figure 7 foods-11-01615-f007:**
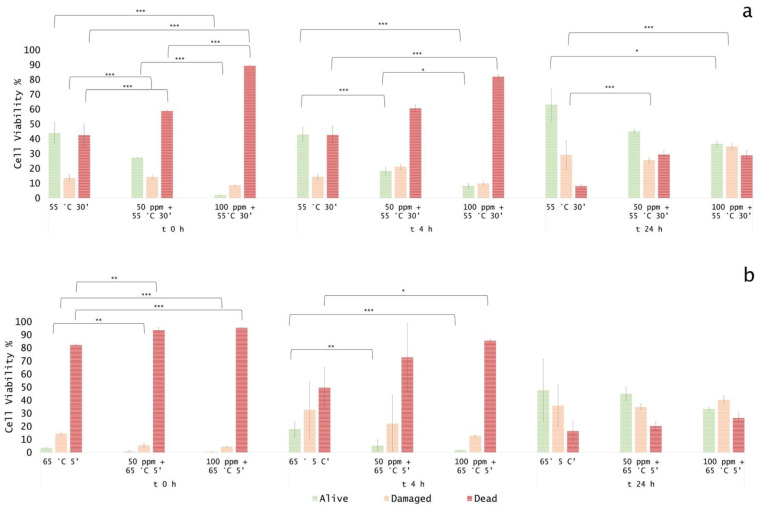
Percentage of alive, dead, and damaged cells calculated on total FCM events (100%): (**a**) combined thermal treatment at 55 °C 30 min and (**b**) 65 °C with or without OEO at 0, 4, and 24 h of incubation, compared to thermal treatment alone. The shown values correspond to the means and standard deviations for replicates, independently analysed. *P*-value less than 0.05 is flagged with one star (*), less than 0.01 with 2 stars (**) and less than 0.001 with three stars (***).

**Figure 8 foods-11-01615-f008:**
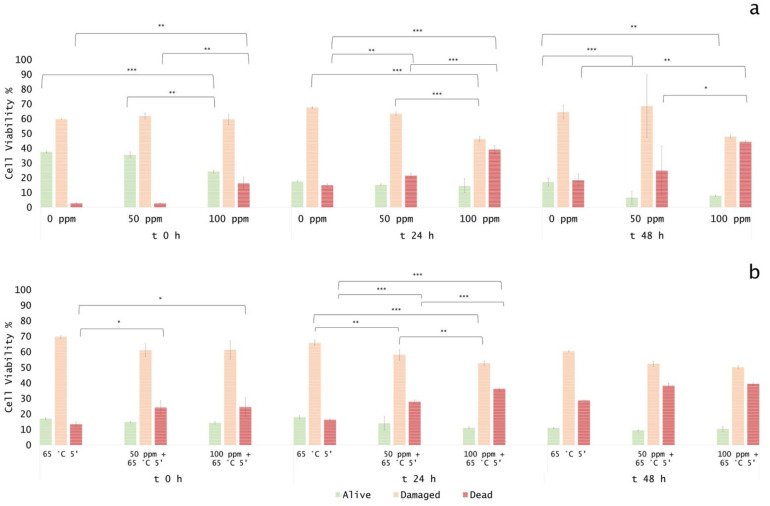
Percentage of alive, dead, and damaged cells concerning total FCM events (100%) in Fruit Juice (pH 3.8): (**a**) OEO treatment and (**b**) combined treatment at 65 °C 5 min. The shown values correspond to the means and respective standard deviations of replicates, independently analyzed. *p*-value less than 0.05 is flagged with one star (*), less than 0.01 with 2 stars (**) and less than 0.001 with three stars (***).

**Figure 9 foods-11-01615-f009:**
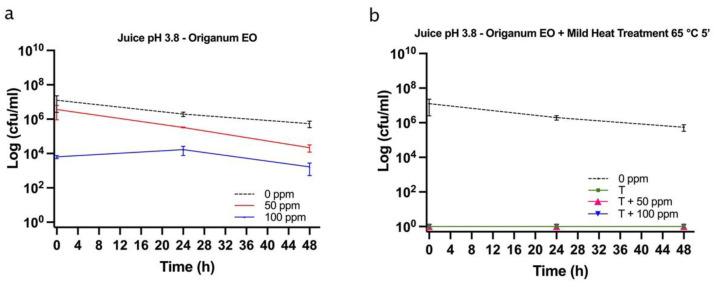
*E. coli* cell load in fruit juice of pH 3.8 plated on MacConkey agar (Log CFU/mL). (**a**) OEO (0, 50, and 100 ppm) treatment and (**b**) thermal treatment 65 °C 5 min alone and combined with OEO (50 and 100 ppm).

**Figure 10 foods-11-01615-f010:**
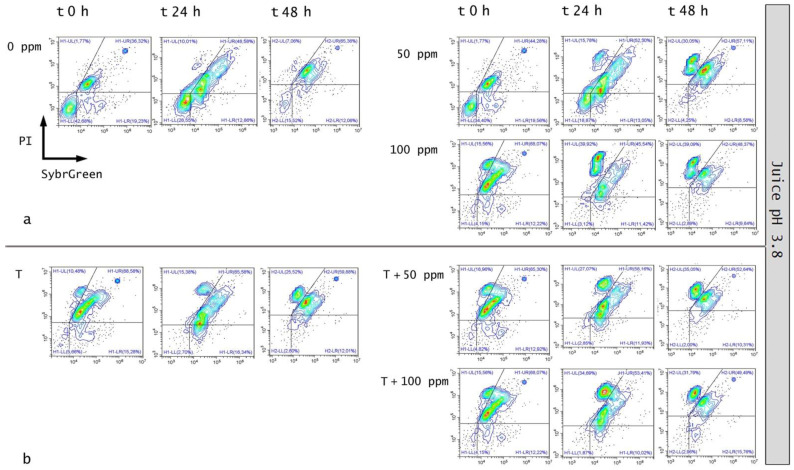
Double-staining density dot plot of *E. coli* cells in fruit juice (pH 3.8) diluted in PBS (pH 7.4): (**a**) 50 ppm and 100 ppm essential oil treatment and (**b**) combined thermal treatment at 65 °C with or without OEO at 0, 24 and 48 h of incubation. Cells were stained with SYBR-Green I and PI simultaneously. H1-LL: unstained debris; H1-LR: intact cells/viable cells; H1-UR: injured cell population; H1-UL: permeabilized/dead cells. In the upper right side of the plot, a defined single population represents standard beads.

**Figure 11 foods-11-01615-f011:**
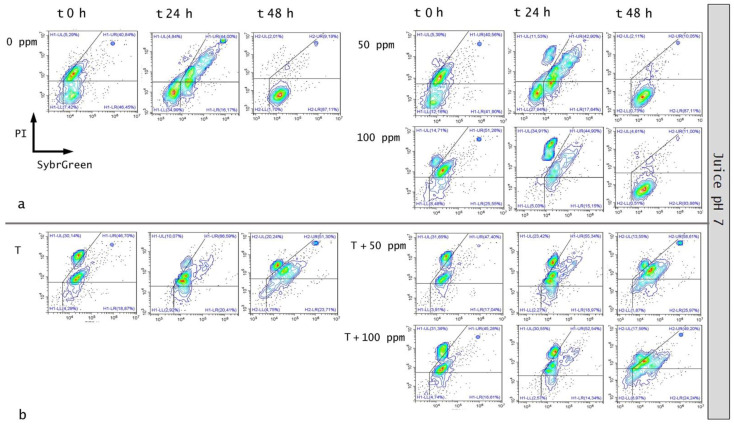
Double-staining density dot plot of *E. coli* cells in fruit juice (pH 7) diluted in PBS (pH 7.4): (**a**) 50 ppm and 100 ppm essential oil treatment and (**b**) combined thermal treatment at 65 °C with or without aroma compounds at 0, 24 and 48 h of incubation. Cells were stained with SYBR-Green I and PI simultaneously. H1-LL: unstained debris; H1-LR: intact cells/viable cells; H1-UR: injured cell population; H1-UL: permeabilized/dead cells. In the upper right side of the plot, a defined single population represents standard beads.

**Figure 12 foods-11-01615-f012:**
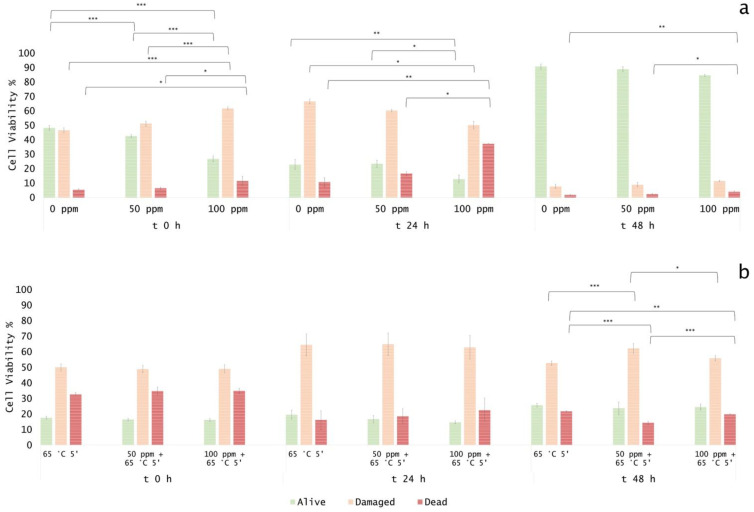
Percentage of alive, dead, and damaged cells calculated on total FCM events (100%) in fruit juice (pH 7). (**a**) EO treatment and (**b**) combined treatment at 65 °C 5 min. The shown values correspond to the means and respective standard deviations of replicates, independently analyzed. *p*-value less than 0.05 is flagged with one star (*), less than 0.01 with 2 stars (**) and less than 0.001 with three stars (***).

**Figure 13 foods-11-01615-f013:**
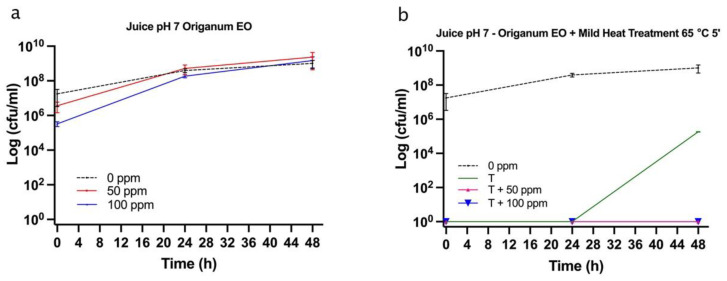
*E. coli* cell load in fruit juice pH 7 plated on MacConkey agar (Log CFU/mL). (**a**) OEO (0, 50, and 100 ppm) treatment and (**b**) thermal treatment 65 °C 5 min alone and combined with OEO (50 and 100 ppm).

**Table 1 foods-11-01615-t001:** Estimation of *E. coli* ATCC 25922 growth by using culture-based method (Log CFU/mL) and population distribution by FCM analysis (Log AFU/mL).

Treatment	Culture-Based Method			Flow Cytometry (% of Events) *			Equivalent Alive Cell Content (Log AFU/mL) **
	Log CFU/mL	Eq Log Reduction	Eq % of Dead Ells	Alive	Damaged	Dead	
Untreated sample	7.44 ± 0.23 ^a^	0	0	98.90 ± 0.13	0.30 ± 0.08	0.80 ± 0.06	7.23 ± 0.02 ^a^
55 °C-30 min	4.99 ± 0.90 ^a^	−2.45 ± 0.90	98.68 ± 2.07	78.06 ± 0.88	18.37 ± 0.88	3.57 ± 0.09	7.05 ± 0.01 ^b^
60 °C-30 min	3.96 ± 0.82 ^a^	−3.48 ± 0.82	99.60 ± 0.16	33.14 ± 0.27	46.42 ± 0.60	20.44 ± 0.40	6.72 ± 0.01 ^b^
65 °C-30 min	1.33 ± 1.24 ^a^	−6.11 ± 1.24	99.99 ± 0.001	1.15 ± 0.18	34.33 ± 3.51	64.59 ± 3.43	5.25 ± 0.05 ^b^

* The percentage presented excludes the background signal obtained with PBS alone on the plots. ** FCM cell content is expressed in Log AFU/mL (Active Fluorescent Units). Within each row, different letters indicate significant differences at *p* < 0.05.

## Data Availability

Not applicable.

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
