# Peer review of "Synergistic Action of Mild Heat and Essential Oil Treatments on Culturability and Viability of Escherichia coli ATCC 25922 Tested In Vitro and in Fruit Juice"

_foods, 2022, doi:10.3390/foods11111615_

Round 1

Reviewer 1 Report

Dear authors,

The article is focused on the evaluation of the antibacterial combination effect between essential oregano oil and mild heating of fruit juices artificially contaminated with Escherichia coli. The evaluation was carried out using a classical methodoly (CFU on agar plates) and determination of the ratio live/damaged/dead cells by using flow cytometry after staining with SYBR-Green I and PI (propidium iodide). The presented investigation is well designed and development of such combined approaches for evaluation of bacterial contamination is very valuable regarding refinement the accuracy of the obtained results and detecting viable but noncutlurable cells.

The English of the article needs refinement.

I have also the following questions?

Point 1: Is the OEO standardized regarding the content of carvacrol or thymol and do the authors have experimental experience with OEO produced by different companies and could there be a difference in the antibacterial effect of the combinations and applied concentrations depending on the content of carvacrol or thymol?

Point 2: It should be valuable to test the effect of the combination heating:OEO on more than one E. coli strain as far as some strains are known to be characterized by a higher thermotolerance than others and are inactivated by temperature above 65 °C. In such cases the addition of OEO could probably contribute also to a better result. Do the authors have experimental experience with such strains?

Point 3: Lines 233 and 234: “Considering the 60 °C and 65 °C heat treatments, an increase of injured cells was observed (46% and 34%, respectively), suggesting a later potential cell recovery.” The sentence is not clear. The comparison group must be specified in order to avoid misunderstanding and false comparison with the temperature values.

Kind regards

Reviewer 2 Report

Manuscript "Synergistic action of mild heat and essential oil treatments on culturability and viability of Escherichia coli ATCC 25922 tested in vitro and in fruit juice" presents interesting research results. The microbiological safety of juices is an important aspect in the production and the search for new methods of preserving these products is very important due to the requirements of consumers who want to give up synthetic preservatives.

Detailed comments:

I am asking the authors to think about the keywords - you should not mention the methods used in the research or the collection numbers of microorganisms.

Chapter 2.2 - please specify which fruit was the juice from.

Drawings should be near the paragraph with the description of a given phenomenon.

The drawings require refinement, the intersection of the axes on the line charts in this case should be around 0.0

The discussion of the results is very weak and requires some refinement.

Reviewer 3 Report

The authors of the paper „Synergistic action of mild heat and essential oil treatments on culturability and viability of Escherichia coli ATCC 25922 tested in vitro and in fruit juice“ assess he individual or combined effect of Origanum vulgare Essential Oil (OEO) and mild heat treatment against E. coli, placing particular emphasis on improvment of the treatment against this bacterium in food.

The article is relevant and timely. The paper is straightforward, well-written and well structured. I find particularly interesting the discussion of results and conclusions even if at some points they are not fully supported by the results. Notwithstanding this, I think they are rich in very interesting and useful insights. Methods are appropriate and adequately described.

I only miss a clearer (and short) explanation on selection of ATCC 255922. Not making this clear may lead to confusion at certain points when reading the paper. Figures and tables are complete, informative and clearly presented.

I include below a couple of additional and minor comments which, if addressed, could improve the paper:

table 1 – why some columns have/have not superscript letters?

please, check English before publication of the paper, in some words missing the inital letter

Reviewer 4 Report

Overall

The topic is of interest to the scientific community and is timely. However, the manuscript is written very poorly. The introduction part must be improved. Recent studies focus on fruit juices where the comparison of mild heat treatment to other treatment are missing. Add some previous related reported work and then explain the gap and what your plans to fill it are. And the author double-checked the language, and typos to minimize the errors. The technical quality of the manuscript is fit for publication.

Abstract 

The abstract should be beginning with a sentence about the background of concept and the aims as well as novelty of study should be mentions. Add space in pH7.0 in line 33.   

Introduction

This part must be improved to explain the background. What is already done and why is this study being carried out? Other than that, the novelty of this research is still poorly presented. References style are not in uniform style. Line 52 have references without space, but in 54 these are with space. Whereas in line 74 author used different style for references. The aim of the study does not represent the research gap and reason for present study.

Materials and Methods

The methods are not properly referenced and are not possible to follow. The author should use SI units throughout the manuscript. i.e., line 109 1 ml whereas in line 111 author used 30 mL, similarly in lines 122, 125, 173, 183, 187 and 440 these are some of them but author should check throughout the manuscript.  

Results and discussion

In my opinion, the author should explain their results and findings in more detail and recent references should be added. The overall discussion part lacks the reasoning and comparison with previous reported work. The results and discussion are not enough to prove the conclusion, and I would suggest rewriting the conclusion with clear novelty.
